# Aerodynamic Uncertainty Quantification for Tiltrotor Aircraft

**Ye Yuan** [1] , **Douglas Thomson** [2] **and David Anderson** [2,*]

1   Department of Aerospace Engineering, Swansea University, Wales SA1 8EN, UK; ye.yuan@swansea.ac.uk
2   Division of Autonomous Systems and Connectivity, University of Glasgow, Scotland G12 8QQ, UK; douglas.thomson@glasgow.ac.uk
*   Correspondence: dave.anderson@glasgow.ac.uk

**Abstract:** The tiltrotor has unique flight dynamics due to the aerodynamic interference characteristics. Multiple aerodynamics calculation approaches, such as the CFD method, are utilised to characterise this feature. The calculation process is usually time-consuming, and the obtained results are generally varied from each other. Thus, the uncertainty quantification (UQ) method will be utilised in this research to identify the aerodynamic inaccuracy effect on the handling qualities of the tiltrotor aircraft. The study aims to quantify the influence of the aerodynamic interference on the tiltrotor flight dynamics in different flight states, such as forward speeds and nacelle tilting angles, which can guide the flight dynamics modelling simplification to improve the simulation efficiency. Therefore, uncertainty identification and full factorial numerical integration (FFNI) methods are introduced to scale these aerodynamic uncertainties. The eigenvalue and bandwidth and phase delay requirements are presented as the failure criteria. The UQ calculation indicates that the uncertainties of the aerodynamic calculation significantly affect the handling quality ratings in two flight ranges: the helicopter mode and the conversion and aeroplane modes with higher forward speed (close to the conversion envelope). Furthermore, a sensitivity analysis is performed to identify the mechanism behind these influences. The results demonstrate that aerodynamics affect the pitching attitude, the pitching damping, and the velocity and incidence stability derivatives. However, the effects of the velocity stability and the incidence stability are the reason causing the handling qualities' degradation in the helicopter mode and high-speed mode, respectively.

**Keywords:** tiltrotor; uncertainty quantification; handling qualities; flight dynamics

## 1. Introduction

Tiltrotor aircraft have drawn extensive research attention as they have the capability to combine the advantages of the fixed-wing airplane and the helicopter, making them capable of both hover and high-speed flight.

However, the aerodynamic interference in the tiltrotor aircraft is more complicated compared with conventional helicopters and fixed-wing aircraft. First, the rotor wake influences the aerodynamic characteristics of the fuselage, wing, and tailplane. It would provide additional aerodynamic payloads to the vehicle, especially in the hover and low speed forward flight ranges. Moreover, the wing wake is coupled with the rotor wake, and therefore, alters the dynamic pressure and incidence of the tailplane, affecting overall trim and stability and controllability features. Additionally, the nacelle incidence angle is a critical factor in determining the aerodynamic interference of tiltrotor aircraft. The rotor wake and its resultant aerodynamics increment on other components are considerably changed when the nacelle is tilted forward or backwards. Furthermore, the trim characteristics, including the attitude and the control inputs, are also decided by the nacelle angles, which, in turn, brings additional alterations to the aerodynamics and flight dynamics features of tiltrotor aircraft. Therefore, the aerodynamic interference affects the flight dynamics characteristics and corresponding handling qualities of the tiltrotor aircraft in different

ways, which should be carefully considered in the flight dynamics modelling and handling quality analysis processes.

Researchers have focused on the tiltrotor aerodynamic interference features for decades. During the XV-15 tiltrotor aircraft design process, a series of wind tunnel experiments and flight tests were implemented to evaluate the aerodynamic interference influence on the trim and performance characteristics. Ferguson [1] constructed the aerodynamic interference model of the XV-15 tiltrotor using the fitting method based on these experiments to improve its accuracy in flight simulations and handling quality investigations. Further, some researchers utilised different methodologies to investigate the aerodynamic interference of the tiltrotor aircraft, including the Free-wake method [2], vortex particle method (VPM) [3,4], and Computational Fluid Dynamics (CFD) method [5–8]. However, the obtained aerodynamics results vary, suggesting the difficulties in calculating accurate results for the aerodynamic interferences in the tiltrotor aircraft. Multiple factors determine the aerodynamic characteristics of the tiltrotor. Except for the factors mentioned above, weather conditions, manufacturing errors, and maintenance states would also influence the resultant aerodynamics. Thus, to improve both the accuracy and computational efficiency of modelling and analysis technologies for the tiltrotors, an alternative approach should be implemented to consider the aerodynamic interference in the relevant investigation.

On the other hand, the uncertainty quantification (UQ) method has been steadily developing in recent years [9,10], providing an efficient approach to investigating complicated systems with multiple uncertainties. By giving the uncertainty inputs, defining the propagation process, and setting relevant failure criteria (thresholds), a quantification result can be obtained by this method to illustrate the probability of the system avoiding given failure conditions in the context of the uncertainty effects. The UQ method has been introduced into the rotorcraft aerodynamics analysis [11–13], in which the manufacturing error effects on rotor aerodynamics and performance were investigated and quantified. Furthermore, the idea of the UQ method was also introduced into the rotorcraft component design process to optimise its power consumption at different flight states [14]. Therefore, it is possible to adopt the uncertainty quantification method into the flight dynamics analysis process to assess and quantify aerodynamic uncertainty effects on flight dynamics and the handling quality ratings, enhancing modelling accuracy.

In light of the preceding discussion, this article first describes the uncertainty quantification methodologies, including the uncertainty identification method, the confidence interval determination, and the variance-based sensitivity analysis techniques. Then, the article introduces the tiltrotor flight dynamics model, especially the aerodynamic interference modelling method. The database of the tiltrotor aerodynamic interference is formed using relevant experiments and numerical calculation results. Failure criteria (thresholds) for quantification are set based on the handling quality specification and the flight dynamics characteristics of the baseline tiltrotor. Then, the handing qualities probabilities of the tiltrotor aircraft are calculated under conditions of different forward speeds and nacelle incidence angles to indicate the aerodynamic uncertainty effects across the flight range. The variance-based sensitivity analysis is also performed to demonstrate the impact of different types of aerodynamic interference on the trim, stability derivative, and controllability derivative results.

### 1.1. Uncertainty Quantification and Sensitivity Analysis Methodologies

1.1.1. Aerodynamics Parameter Identification

The output error method is utilised in this section for parameter identification, which is used to determine the uncertainty parameter that approaches the best value estimate as the number of data increase.

For the tiltrotor aircraft and any other rotorcraft, its flight dynamics model can be written as a set of the following non-linear equations considering the effect of uncertainties

$$\dot{x}(t) = f(x(t), u(t), \delta, t) \tag{1}$$

where $x$ and $u$ are the state and control vectors, respectively. $t$ is the response time. $\delta$ is the aerodynamic interference vector for uncertainty quantification analysis. Then, the observed output, $y$, can be represented using observer transformation, $g$, which is

$$y(t) = g(x(t), u(t), \delta) \tag{2}$$

where the observed output in this investigation includes the flight dynamics characteristics of the tiltrotor, such as trim characteristics, stability and controllability derivatives, and the handling quality requirements. The measured data, $z$, should be provided as

$$z(t) = y(t) + v(t) \tag{3}$$

where $v$ denotes the measurement noise. This noise is assumed to be a sequence of independent zero-mean Gaussian random numbers, which can be used to account for the differences between various research works in the tiltrotor aerodynamics calculation.

Thus, the estimates of $\delta$ are found by minimising the cost function [15]:

$$J = \frac{1}{2} \sum_{k=1}^{N} [z_k - y_k]^T R^{-1} [z_k - y_k] \tag{4}$$

where $R$ is the prediction error covariance matrix. $[z_k - y_k]$ denotes the vector of the difference between measured data and calculation results in terms of the $k$-th experiment data. The minimum of the cost function can be calculated using a Gauss–Newton method as follows:

$$\delta_{i+1} = \delta_i + \Delta\delta \tag{5}$$

$$\Delta\delta = -M^{-1}G \tag{6}$$

where $M$ and $G$ represent the information matrix and gradient vector, respectively, and they can be calculated using the following equations:

$$M = \sum_{k=1}^{N} \left[\frac{\partial y_k}{\partial \delta}\right]^T R^{-1} \left[\frac{\partial y_k}{\partial \delta}\right] \tag{7}$$

$$G = \sum_{k=1}^{N} \left[\frac{\partial y_k}{\partial \delta}\right]^T R^{-1} [z_k - y_k] \tag{8}$$

These values require first-order derivatives to calculate the uncertainty results, and consequently, the flight simulation model incorporating the automatic differentiation method can improve the computational efficiency of this identification process. Using the calculation process of Equations (4)–(8), the uncertainty vector $\delta$ can be determined by values that minimise the error between the modelling calculation and measurement results, forming the basis for the uncertainty quantification process.

### 1.1.2. Uncertainty Quantification Node Selection

The Full Factorial Numerical Integration (FFNI) method [16] will be used to calculate the uncertainty quantification determining nodes and the corresponding weights based on the identification results. A brief introduction of the FFNI method will be illustrated.

Using the FFNI method, the statistical moments of the performance indices are calculated through direct numerical integration. In the numerical analysis, a quadrature formula approximates the definite integral of a function, usually expressed as a weighted sum of function values at specified points in the domain of integration. Thus, the $m$-node quadrature formula for statistical moments can be written as

$$E[f^k] = \int \{f(\delta)\} g_p(\delta) d\delta \approx \sum_{i=1}^{m} \omega_i [f(\mu + \alpha_i \sigma)]^k \tag{9}$$

where $g_p$ is the probability density function, and $\alpha_i$, $\omega_i$ are the location parameter of the $i$-th node and the corresponding weight, respectively. Therefore, the optimal locations of the calculation nodes and the corresponding weights can be calculated using the moment-matching equations below

$$M_k = \int (\delta - \mu)^k g_p(\delta)d\delta = \sum_{i=1}^{m} \omega_i(\alpha_i\sigma)^k, k = 0, \ldots, 2m-1 \tag{10}$$

where $M_k$ is the $k$-th central moment of random variable $\delta$ (Aerodynamic uncertainty factors). The non-linear system of equations (Equation (1)) can be solved with numerical methods to find the unique $\{\alpha_1, \ldots, \alpha_m, \omega_1, \ldots, \omega_m\}$.

### 1.1.3. Sensitivity Analysis Algorithm

The variance-based sensitivity analysis method will be introduced to assess the effect of different aerodynamic interference on flight dynamics characteristics. Thus, a brief introduction of the sensitivity analysis method is shown below based on references [17,18].

Based on the FFNI results, Sobol's sequence [19,20] is constructed to represent the uncertainties in the aerodynamics of the tiltrotor, which is used to form an $N \times 2D$ sample matrix, where $N$ and $D = 4$ represent the number of sampling and uncertainty inputs, respectively. The first $D$ columns of the matrix form matrix $A$ and the remaining $D$ columns are regarded as matrix $B$. Then, the matrix $A_B{}^i$ ($i = 1, 2,..., D$) is constructed by replacing the $i$-th column of $A$ with the $i$-th column of matrix $B$. Then, the total effect index, $S_{Ti}$, is calculated using the following equations to reflect the importance of each aerodynamic interference part in determining the flight dynamics characteristics, in which the interaction amongst different uncertainties is also included in this index.

$$S_{Ti} = \frac{E_{X\sim i}(Var_{Xi}(Y/\mathbf{X}_{\sim i}))}{Var(Y)} \tag{11}$$

where:

$$E_{X\sim i}(Var_{Xi}(Y/\mathbf{X}_{\sim i})) \approx \frac{1}{2N}\sum_{j=1}^{N}\left(f_{un}(\mathbf{A})_j - f_{un}(\mathbf{A_B}{}^i)\right)^2 \tag{12}$$

$$Var(Y) = Var\left[\begin{array}{c} f_{un}(\mathbf{A}) \\ f_{un}(\mathbf{B}) \end{array}\right] \tag{13}$$

In Equations (12) and (13), $f_{un}$ denotes the correlation between the uncertainty inputs and the flight dynamics characteristics, such as the trim parameters, stability derivatives, damping derivatives, and controllability derivatives.

## 2. Analysis Methodology

The aerodynamic interference of the tiltrotor aircraft is derived from the rotor wake and wing wake. The rotor wake could add velocities to other components of tiltrotor aircraft, such as the wing and pylon parts and tailplane parts. However, the wake of the wing mainly changes the airflow direction when it arrives at the tailplane region. Thus, the wake interference would alter the incidence angle of the tailplane, resulting in the change of its resultant forces and moments. Therefore, multiple aerodynamic interferences influence the flight dynamics characteristics of the tiltrotor in various ways, and a physics-based analytical model that reflects these interferences directly can be helpful to observe the relationship between aerodynamic interference and flight dynamics characteristics of the tiltrotor.

### 2.1. Tiltrotor Flight Dynamics Model

The flight dynamics model is the basis to perform the aerodynamic uncertainty quantification and sensitivity analysis. The developed tiltrotor flight dynamics model is introduced here, and its detail and validation process can be found in reference [17]. The

incorporated automatic differentiation algorithm will help to reduce the computational expense of the uncertainty propagation process, especially the differentiation process (Equations (7) and (8)) in the uncertainty identification.

More importantly, this flight dynamics model can provide a physics-based modelling method to analyse the tiltrotor aerodynamic interference phenomenon, which uses four parameters, $\delta_{rw}$, $\delta_{rh}$, $\delta_{rv}$, and $\delta_{wt}$ to denote the aerodynamic interference factors between rotor and wing, rotor and horizontal tail, rotor and vertical tail, and wing and horizontal tail, respectively. The effect of these parameters on tiltrotor flight dynamics can be represented as follows, which is formed based on the projection relationship between the rotor and wing wake and corresponding components:

$$\begin{bmatrix} X_{w,\text{int}} \\ Y_{w,\text{int}} \\ Z_{w,\text{int}} \end{bmatrix} = \begin{bmatrix} \delta_{rw} v_i f_w(\beta_m) \\ 0 \\ \delta_{rw} v_i g_w(\beta_m) \end{bmatrix} \tag{14}$$

$$\begin{bmatrix} U_{ht,\text{int}} \\ V_{ht,\text{int}} \\ W_{ht,\text{int}} \end{bmatrix} = \begin{bmatrix} \delta_{rh} v_i \sin(\beta_m) \\ 0 \\ -\delta_{rh} v_i \cos(\beta_m) \end{bmatrix} \tag{15}$$

$$\begin{bmatrix} U_{vt,\text{int}} \\ V_{vt,\text{int}} \\ W_{vt,\text{int}} \end{bmatrix} = \begin{bmatrix} \delta_{rv} v_i \sin(\beta_m) \\ 0 \\ -\delta_{rv} v_i \cos(\beta_m) \end{bmatrix} \tag{16}$$

$$\alpha_{ht,\text{int}} = \delta_{wt} \tag{17}$$

where $v_i$ and $\beta_m$ are average induced velocities on the rotor disc and the nacelle incidence angle, respectively. According to the relevant modelling report and wind tunnel experiments [1,21,22], the wing-vertical tail interference is ignored in the tiltrotor aircraft as their relative position suggests that this interference cannot alter the overall flight dynamics to a large extent. The analytical method shown in Equations (14)–(17) provides physics meanings of the aerodynamic interference effects as they are obtained from the fixed-wake theory [1]. The uncertainty quantification results derived from this research would indicate the interference effect on the flight dynamics and provide quantitative information on the internal mechanisms of these interferences.

Therefore, considering the aerodynamic interference effect, the tiltrotor flight dynamics model can be represented as a set of non-linear differential equations shown in Equation (1). The state vector, x, contains the angular velocities, blade dynamics motions, and induced velocities. The control vector, *u*, includes the collective pitch, longitudinal and lateral controllers, and pedal input. *t* is the response time. $\delta = [\delta_{rw}, \delta_{rh}, \delta_{rv}, \delta_{wt}]$ is the aerodynamic interference vector for uncertainty quantification analysis.

Additionally, the control strategy of this model follows relevant information from reference [1]. The dynamic models of the controller and actuator are added to the flight dynamics model to calculate the bandwidth and phase delay results with more accuracy. According to the relevant article [23,24], corresponding transfer functions are shown below:

$$S_{Control} = \frac{16.9747}{s^2 + 44.4s + 986} \tag{18}$$

$$S_{Actuator} = \frac{1}{0.02s + 1} \tag{19}$$

where $S_{control}$ and $S_{Actuator}$ are the dynamic models of the control mechanisms and actuators, respectively.

### 2.2. Aerodynamic Uncertainty Inputs

The confidence interval ranges of different elements in $\delta$ should be determined using Equations (4)–(8), and the FFNI method will form uncertainty inputs for the quantification

propagation. A series of aerodynamic interference research results related to the tiltrotor aircraft will be used as a database to determine the uncertainty interval range, which is shown in Table 1.

**Table 1.** Aerodynamic interference calculation references.

| Reference | Interference Type | Flight Range | Research Method | Weight |
|---|---|---|---|---|
| [25] | Rotor and Wing/Rotor and Tail/Wing and Tail | All | Wind tunnel experiments | 1.0 |
| [22] | Wing and Tail | Helicopter mode | Wind tunnel experiments | 1.0 |
| [5] | Rotor and Wing | All | CFD calculation | 0.5 |
| [26] | Rotor and Wing | Hover | Wind tunnel experiment | 1.0 |
| [6] | Rotor and Wing | Conversion mode | CFD calculation | 0.5 |
| [7] | Rotor and Wing | Helicopter mode | CFD calculation | 0.5 |

The data from Table 1 are introduced into the uncertainty identification process to calculate its corresponding uncertainty factor $\delta$. Additionally, it should be mentioned that the confidence weight (the last column in Table 1) is based on the research method utilised (wind tunnel experiment corresponds to 1.0, and the CFD calculation corresponds to 0.5).

### 2.3. Threshold for Quantification

The eigenvalue and bandwidth and phase delay requirements from rotorcraft handling qualities specification [27] will be used in this investigation as quantification thresholds, shown in Figures 1 and 2, respectively. Further, the calculation results from the baseline tiltrotor are also added to these figures.

As illustrated in Figure 1, the eigenvalue results are varied along with forward speed and nacelle incidence angle. In hover and low speed forward flight, the eigenvalue ratings are in Level 2 or Level 3, and it is improved with forward speed and nacelle angle. This is because the stability moment of tailplanes becomes more significant in the high-speed range.

Figure 2 indicates that the forward speed and nacelle incidence angle influence the bandwidth and phase delay characteristics. Overall, both longitudinal and lateral bandwidth results are improved along with forward speed, and the phase delay roughly remains at 0.05 s across the flight range. The forward speed growth provides additional damping and stability derivatives in different channels, resulting in improved bandwidth rating.

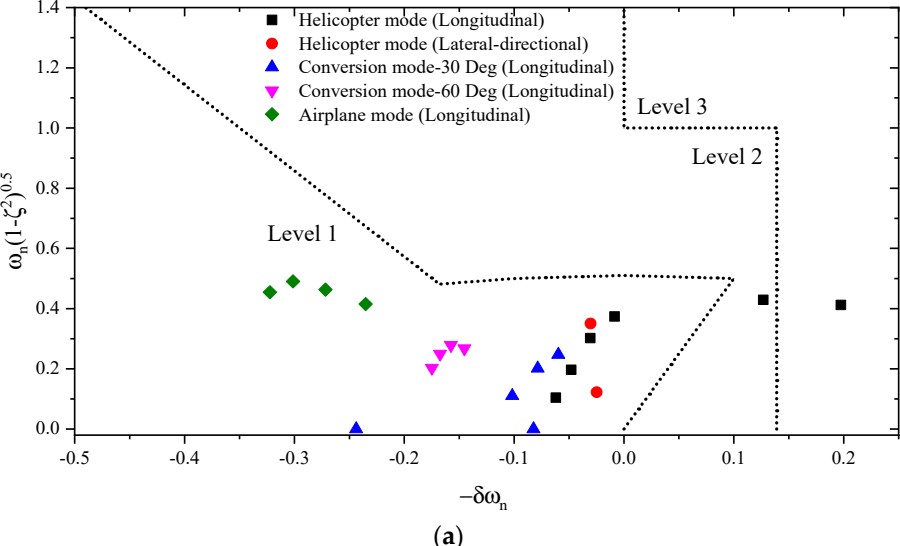

(**a**)

**Figure 1.** *Cont.*

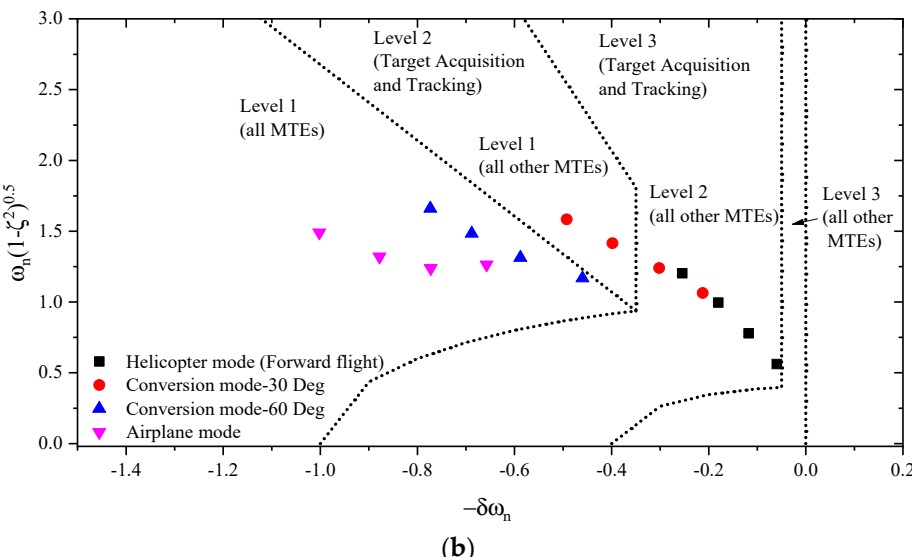

(**b**)

**Figure 1.** The eigenvalue requirements in different flight ranges. (**a**) Longitudinal requirement and lateral-directional requirement in hover and low-speed forward flight; (**b**) Lateral-direction oscillatory mode requirement.

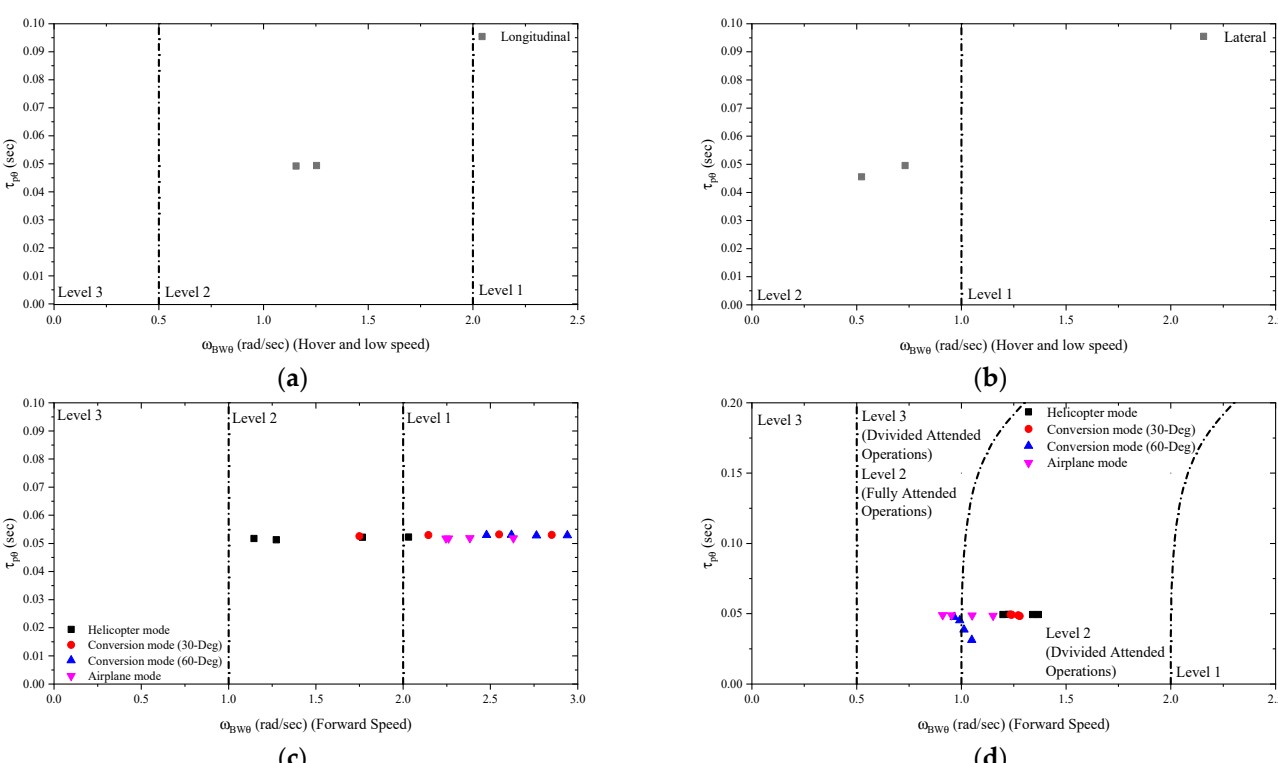

**Figure 2.** The bandwidth and phase delay requirements in different flight ranges. (**a**) Longitudinal-Hover and low-speed flight; (**b**) Lateral-Hover and low-speed flight; (**c**) Longitudinal-forward flight; (**d**) Lateral-forward flight.

Based on the calculations, the eigenvalue and bandwidth and phase delay thresholds in the UQ process are set based on the nacelle incidence angle, as shown in Tables 2 and 3, respectively. The safety conversion envelope of the tiltrotor aircraft implies that the forward speed influence is also included in these criteria.

**Table 2.** Eigenvalue thresholds for uncertainty quantification.

| $B_m$ (Deg) | Longitudinal Threshold | Lateral-Directional Threshold |
|---|---|---|
| $0 < \beta_m \le 15$ | Level 2–Level 3 | Level 2–Level 3 (All other MTEs) |
| $15 < \beta_m \le 30$ | | Level 1–Level 2 (All other MTEs) |
| $30 < \beta_m \le 60$ | Level 1–Level 2 | |
| $60 < \beta_m \le 75$ | | Level 1–Level 2 (Target Acquisition and Tracking) |

**Table 3.** Bandwidth and phase delay thresholds for uncertainty quantification.

| $B_m$ (Deg) | Longitudinal Threshold | Lateral-Directional Threshold |
|---|---|---|
| $0 < \beta_m \le 15$ | Level 2–Level 3 | |
| $15 < \beta_m \le 30$ | | Level 3–Level 2 (Fully Attended Operation) |
| $30 < \beta_m \le 60$ | Level 1–Level 2 | |
| $60 < \beta_m \le 75$ | | |
| $75 < \beta_m \le 90$ | | |

The UQ threshold considers the performance characteristics and the handling qualities of the baseline calculation results. Firstly, the tiltrotor aircraft is usually evaluated based on the utility or cargo rotorcraft requirements in the rotorcraft handling quality specification, i.e., this rotorcraft configuration needs less agility requirement. Besides that, these thresholds should not be far away from the baseline results so that the aerodynamic uncertainty effect can be observed to a greater extent. Therefore, the ADS-33 is utilised here as the basis for deciding the threshold for the UQ process. On the other hand, the handling qualities are also determined by the flight speed and nacelle incidence angle. Thus, with the aim of demonstrating the aerodynamics effect more accurately, we alter the threshold through the level defined in the ADS-33 along with the flight states.

It should be mentioned that, according to the rotorcraft handling quality specification, the relevant requirements are different in hover and low-speed flight (below 45 knots) and forward flight (above 45 knots), which may lead to an additional alteration in the probability calculations.

## 3. Quantification Results

The handling quality quantification results with different nacelle incidence angles and forward speeds are shown in Figure 3, and separate figures of eigenvalue and bandwidth and phase delay probability results are also added here.

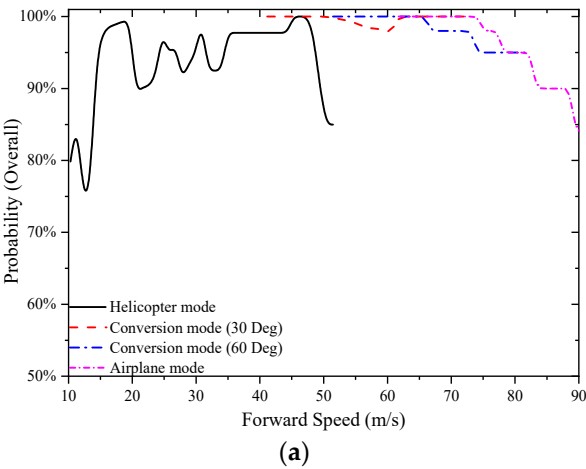

(**a**)

**Figure 3.** *Cont.*

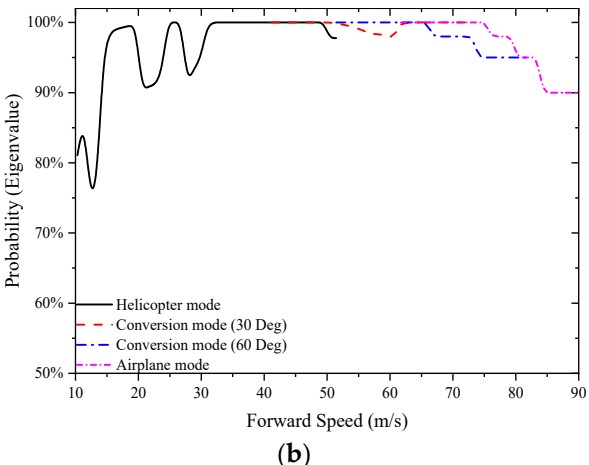
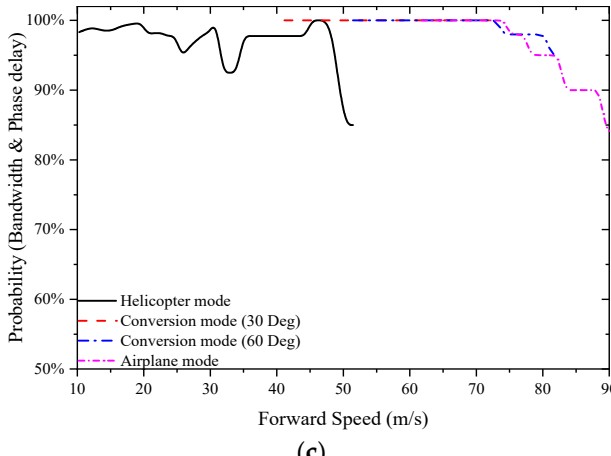

**Figure 3.** Aerodynamic uncertainty quantification results. (**a**) Overall probability results; (**b**) Eigenvalue probability results; (**c**) Bandwidth and Phase delay probability results.

As indicated in Figure 3, the aerodynamic interference influences the eigenvalue and bandwidth and phase delay results in the helicopter mode and the conversion and airplane mode with higher forward speed. Thus, a simplified aerodynamic modelling method can be introduced to tackle the aerodynamic modelling method in other flight ranges to improve the simulation efficiency.

The aerodynamic uncertainty alters both the eigenvalue and bandwidth and phase delay rating results in the helicopter mode. The eigenvalue probabilities are changed dramatically between 75% to 100% in low-speed and mid-speed flight ranges. Furthermore, significant oscillations in the bandwidth and phase delay probability result in a higher forward speed in this mode. As analysed later in this article, the aerodynamic interference is complicated in the helicopter mode. When the tiltrotor is in hover or low speed forward flight, the rotor and wing interference would noticeably increase the payload of the wing and pylon part. Then, by increasing the forward speed, the projection positions of the rotor wake on the wing surface are moving backwards quickly, and this phenomenon is diminished. However, the aerodynamic interferences would take more effects with speed increases, resulting in an extra alteration of the handling quality probability results. Besides, the handling quality requirements are altered from the low-speed criteria to forward flight criteria when the flight speed is 45 knots, which also influences the probability results.

When the tiltrotor aircraft is in the conversion mode, the aerodynamic uncertainty on the probability is reduced. Based on Figure 3, the overall quantification results are above 94% across the conversion mode. The aerodynamic uncertainty would still affect the flight dynamics during the conversion mode. However, as the handling quality ratings are improved, these effects are hard, to drive a degradation in the handling quality ratings.

The aerodynamic uncertainty still alters the tiltrotor handling qualities in airplane mode. According to the analysis later in this article, only wing and tailplane interference can significantly affect the flight dynamics characteristics in airplane mode. However, this interference can still degrade the handling qualities at the high-speed range in this mode. Both eigenvalue and bandwidth and phase delay ratings are reduced in the high-speed range (above 70 m/s), as shown in Figure 3.

## 4. Sensitivity Analysis

The quantification results demonstrate that aerodynamic interference plays a considerable effect on the handling qualities of the tiltrotor aircraft. With the aim to investigate this effect, sensitivity analysis is performed to assess the correlation between different types of aerodynamic interference and the flight dynamics features, including trim, stability, and controllability characteristics.

### 4.1. Trim Analysis

The effect of aerodynamic interference on the trim characteristics in different nacelle incidence angles and the corresponding total effect index results are shown in Figure 4.

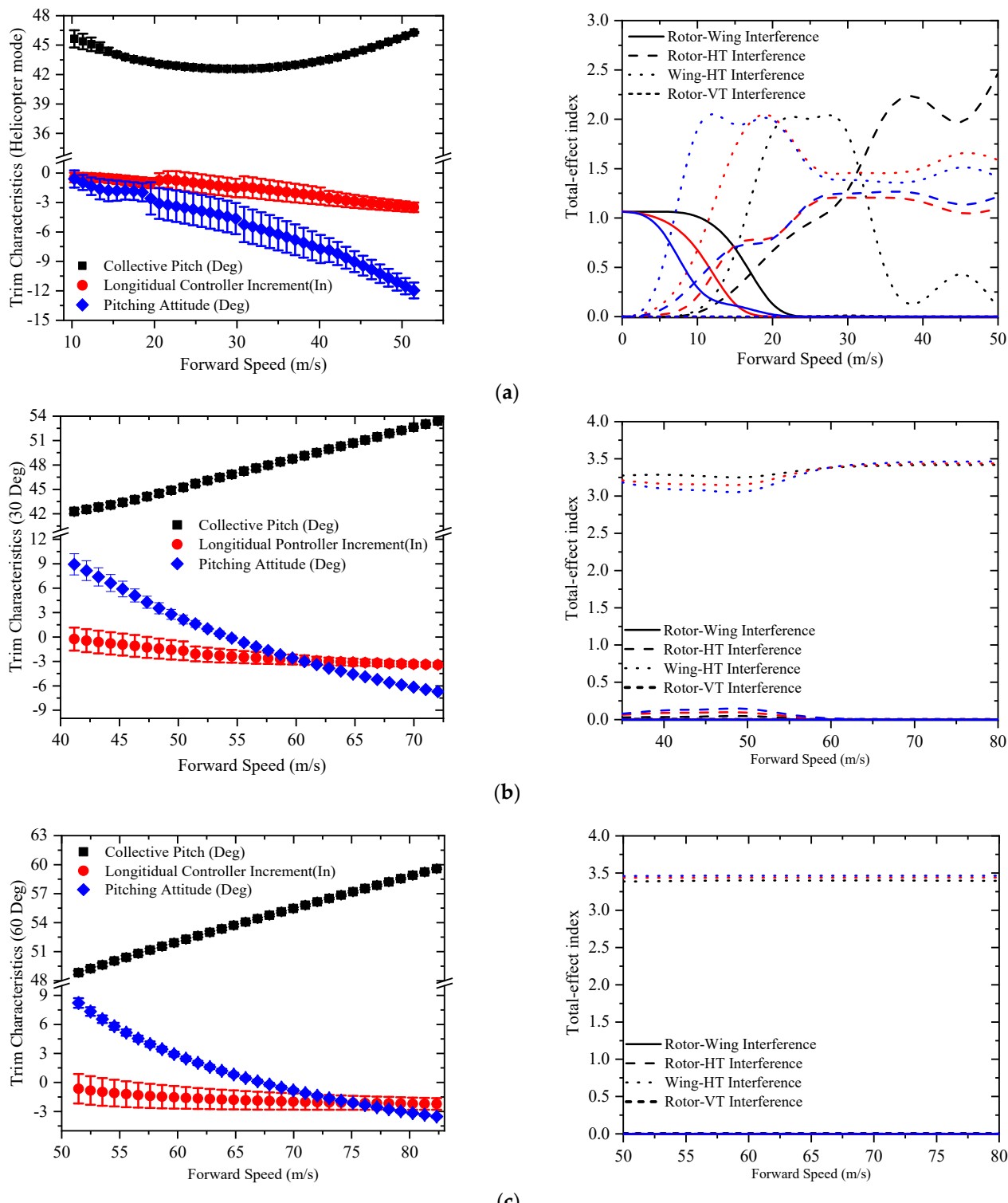

**Figure 4.** *Cont.*

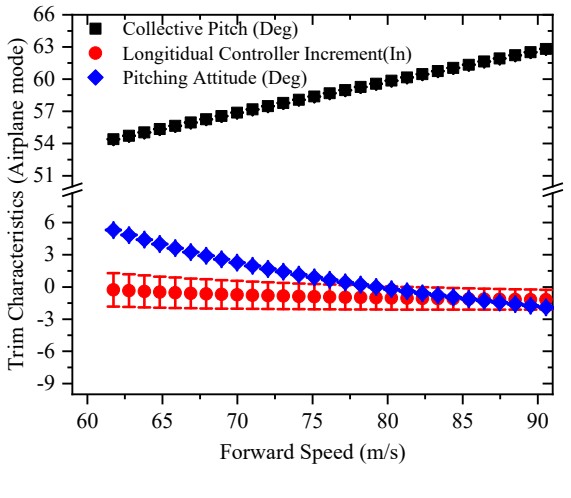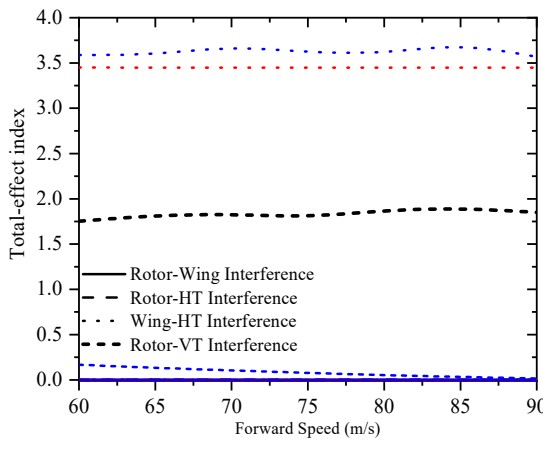

**(d)**

**Figure 4.** Trim sensitivity results. (**a**) Helicopter mode; (**b**) Conversion mode (30 Deg); (**c**) Conversion mode (60 Deg); (**d**) Airplane mode.

Figure 4 indicates that the aerodynamic interference alters the trim characteristics. In the helicopter mode, the rotor-wing interference exaggerates the load of the wing part and consequently changes the collective pitch in the low-speed forward flight. With the increase of the forward speed, its influence on the wing component is diminished. Therefore, the impact on the collective pitch is reduced as the forward speed increases. Although other aerodynamic interferences still affect the collective pitch results, the magnitude of those influences is much lower than that derived from the rotor-wing interference.

On the other hand, the longitudinal controller and pitching attitude are significantly affected by aerodynamic interferences. The wake affects the aerodynamic characteristics of the horizontal tail, and consequently, an additional pitching moment is produced. This moment increment would alter the longitudinal trim result. Besides, it should be mentioned that when the tiltrotor is in helicopter mode with a higher forward speed (50 m/s, for example), the pitching attitude exceeds −12 Deg, causing the attack angle of the horizontal tail to close or even exceed the stalling limit. Thus, the aerodynamic interference on the horizontal tail may create a significant non-linearity, deteriorating the handling qualities at this flight range.

The aerodynamic interference changes the longitudinal control input across the flight range in the conversion and aeroplane mode. However, its effect on the pitching attitude is diminished as the forward speed and nacelle incidence angle increase. The induced velocity of the rotor disc decreases along with the forward speed and nacelle incidence, and consequently, the rotor wake influence on the wing and tail surface parts is reduced, which can be observed in the sensitivity analysis results. However, the wing-horizontal tail interference still affects the trim characteristics, and the interference leads to a significant increment of the attack angle on the tailplane. As the elevator provides most of the pitching control power in the conversion and airplane mode [1], the longitudinal controller is capable of compensating for this influence without altering the pitching attitude to a large extent. Besides, it should be mentioned that the rotor-vertical tail is the key factor in determining the collective pitch trim results in the aeroplane mode. The collective pitch provides the forwarding force, rather than the vertical force, in the aeroplane mode, and the projection relationship between the rotor disc and vertical tails indicates that this interference produces additional drag that needs extra collective pitch. However, although the rotor-vertical tail interference is the key player in changing the collective pitch in the aeroplane mode, this influence is still minimum and can hardly influence the trim results.

### 4.2. Control Derivative Analysis

The on-axis control derivatives ($\delta q/\delta X_{lon}$, $\delta p/\delta X_{lat}$) with different nacelle incidence angles and corresponding total effect index results are shown in Figure 5.

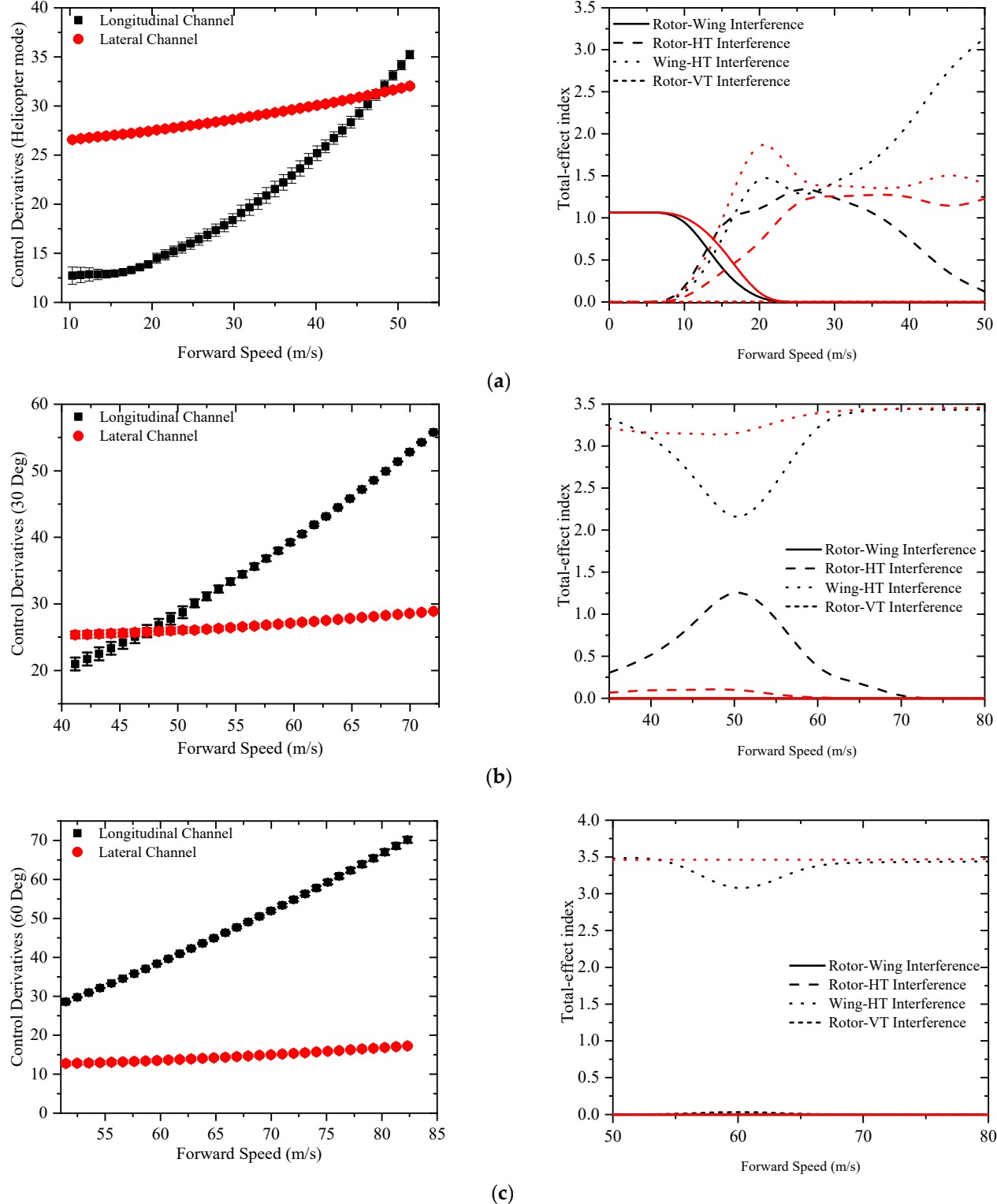

**Figure 5.** *Cont.*

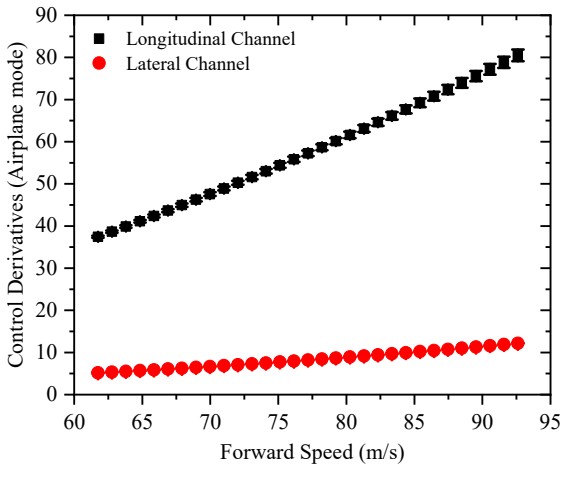 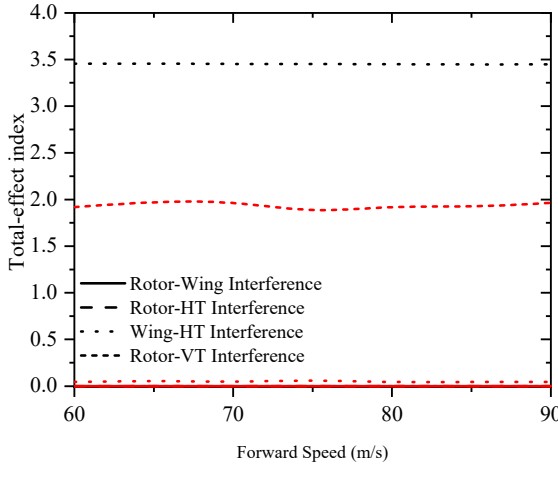

**(d)**

**Figure 5.** On-axis control derivative results. (**a**) Helicopter mode; (**b**) Conversion mode (30 Deg); (**c**) Conversion mode (60 Deg); (**d**) Aeroplane mode.

According to Figure 5, the aerodynamic uncertainty can affect the longitudinal on-axis control derivatives, and the effect on the lateral control derivatives is much smaller and reduced as forward speed increases.

In the longitudinal channel, the aerodynamic interferences of the rotor wake and wing wake mainly influence the attack angle and dynamic pressure on the tail surface. As indicated in Figure 5, the longitudinal on-axis control derivatives are changed in the helicopter mode. With forward speed and nacelle incidence angle increases, the rotor-horizontal tail interference influence on the longitudinal control power is reduced as the wake is skewed backwards, and only the wing wake can change the longitudinal control derivatives in the conversion and airplane mode.

The interference could only affect the lateral control power by altering the dynamic pressure of the aileron region. When the tiltrotor aircraft is in hover or low-speed flight, the rotor-wing aerodynamic interference is relatively more significant, leading to a noticeable change in lateral control power. When the tiltrotor aircraft is in high-speed flight and/or in conversion and airplane mode, the rotor wake is tilted backwards, making the aerodynamic interference's influence on the aileron part diminished. The other aerodynamic interferences can indirectly alter the lateral control power by influencing the rotor aerodynamics and corresponding trim state. However, this influence is small across the flight range, as indicated in Figure 5. Additionally, it should be mentioned that the decrease of the lateral on-axis control derivative along with the nacelle incidence angle is because of the control allocation between the differential collective and the aileron deflection.

*4.3. Stability Derivative Analysis*

The velocity, incidence, dihedral-effect, and yawing stability derivative results in different nacelle incidence angles and corresponding total effect index results are shown in Figure 6.

Figure 6 indicates that the aerodynamic interference affects the longitudinal stability derivatives, namely, the velocity stability derivative ($\delta q / \delta V_x$) and the incidence stability derivative ($\delta q / \delta V_z$). The interference effect on dihedral-effect stability ($\delta p / \delta V_y$) and heading stability derivative ($\delta r / \delta V_y$) is much lower. The tiltrotor aircraft configuration is symmetric, and the aerodynamic uncertainty effects are mainly related to the longitudinal and vertical channel. Consequently, its influence on the lateral/yawing channel is reduced.

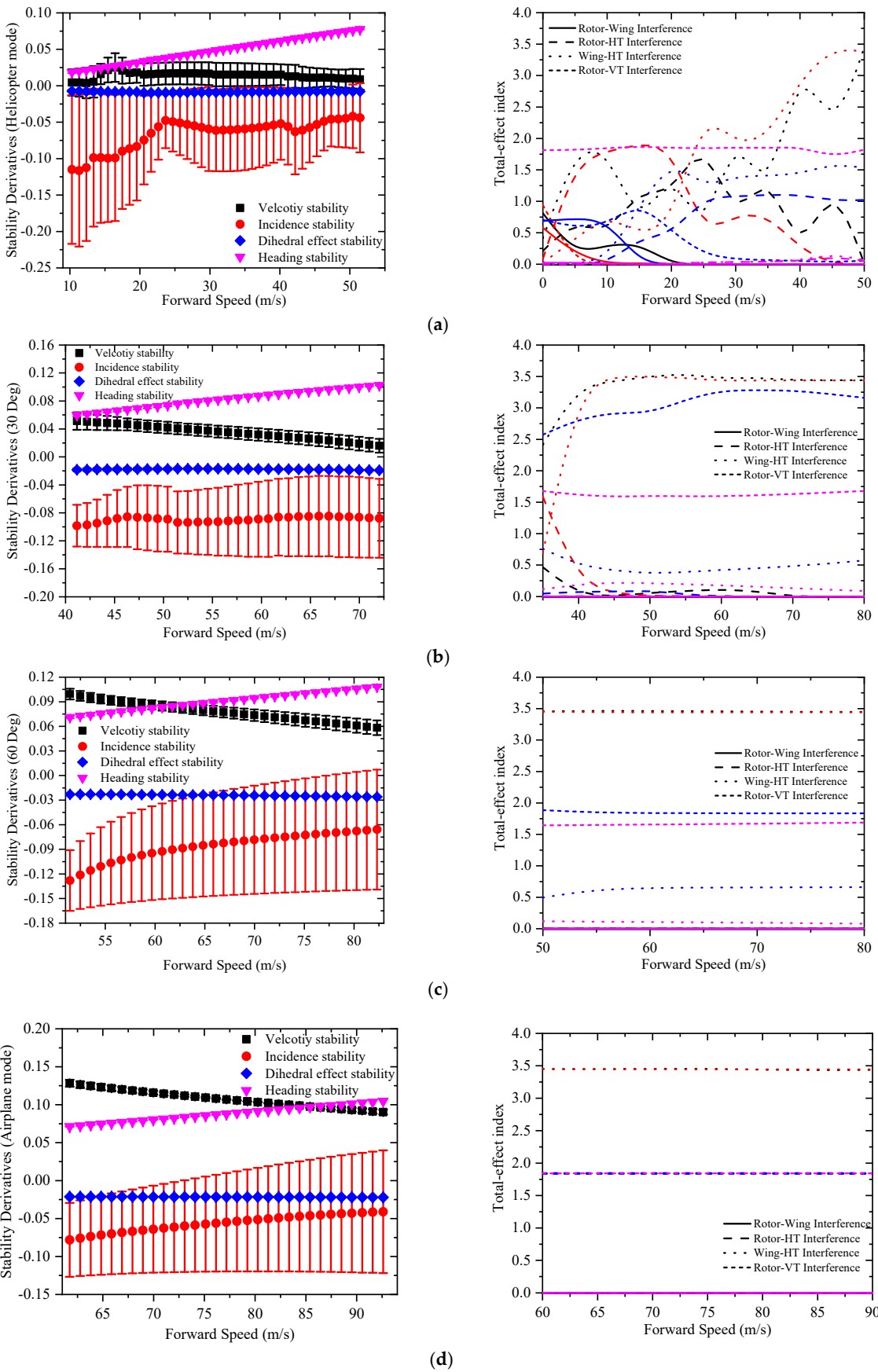

**Figure 6.** Stability derivative results. (**a**) Helicopter mode; (**b**) Conversion mode (30 Deg); (**c**) Conversion mode (60 Deg); (**d**) Airplane mode.

The effect of the aerodynamic interference on the velocity stability derivatives is most significant in the helicopter mode. During this flight range, the aerodynamic interferences, especially the rotor-horizontal tail interference and the wing-horizontal tail interference, alter the pitching moment provided by the wing and tailplane parts. Furthermore, as the velocity stability is close to zero in the helicopter mode, the aerodynamic interference may lead this stability derivative to be negative, resulting in the degradation in the handling qualities ratings. Compared to the uncertainty quantification results from Figure 3, the probability results decrease as the velocity stability range is close to zero, demonstrating a straightforward relationship between the interference-induced velocity stability increment and the handling quality rating. When the pylon is tilted forward and forward speed increases, the velocity stability is improved. Therefore, only the wing-horizontal tail interference can alter the velocity stability derivative in the conversion and aeroplane mode, and the corresponding influential magnitude is much lower than that in the helicopter mode.

The aerodynamic interference alters the incidence stability to a large extent across the flight range. The wakes of the rotor and wing parts change the dynamic pressure and attack angle of various components in the vehicle. In the helicopter mode and the conversion mode with a lower nacelle angle, both rotor and wing wakes would influence the aerodynamics characteristics of the tailplane together. Therefore, a non-linear correlation between the uncertainty-induced incidence stability increment and forward speed can be observed because of the coupling effect between these two wakes. When the nacelle incidence changes to 60 Deg, only the wing and tailplane interference can be observed in terms of the incidence stability, and the corresponding uncertainty effect on incidence stability grows with the forward speed. Based on Figure 6c, d, the incidence stability results are close to zero in high-speed flight, leading the handling qualities ratings to be degraded, as shown in Figure 3.

The slight influence on lateral/yawing stability is mainly due to the rotor-vertical tail interference. As shown in Figure 6, this influence cannot introduce a significant change in the relevant stability derivatives and consequently cannot affect the overall handling qualities. The sensitivity index also demonstrates indirect correlations between other interferences and the lateral stability derivatives, especially during the helicopter mode. These interferences can alter the trim characteristics, such as the pitching attitude, resulting in a slight change in the lateral stability.

### 4.4. Angular Damping Derivative Analysis

Figure 7 shows the angular velocity damping derivatives and corresponding total effect indices in different nacelle incidence angles.

As indicated in Figure 7, the aerodynamic uncertainty influences the pitching damping derivatives across the flight range. The pitching damping is mainly affected by the rotor-horizontal tail interference and the wing-horizontal tail interference, in which the rotor-horizontal tail interference is more significant in helicopter mode, and the wing-horizontal interference is the influential factor across the flight range. Furthermore, the aerodynamic uncertainties slightly affect the rolling and yawing damping derivatives. The rotor-vertical tail interference alters the dynamic pressure of the vertical tail surface and provides additional rolling and yawing damping derivatives. Other interference may change these damping derivatives slightly by altering the overall trim state at the given flight range.

It should be mentioned that the pitching damping derivative is relatively small in low-speed forward flight. This is because the gimbal rotor system is adopted in tiltrotor aircraft to overcome the aeroelastic instability [28]. However, the pitching damping provided by this rotor hub system is much lower. The reduction of the pitching damping may lead to the aerodynamic uncertainty effect on handling quality ratings being more significant, which can be reflected by the probability results in Figure 3 in the helicopter mode.

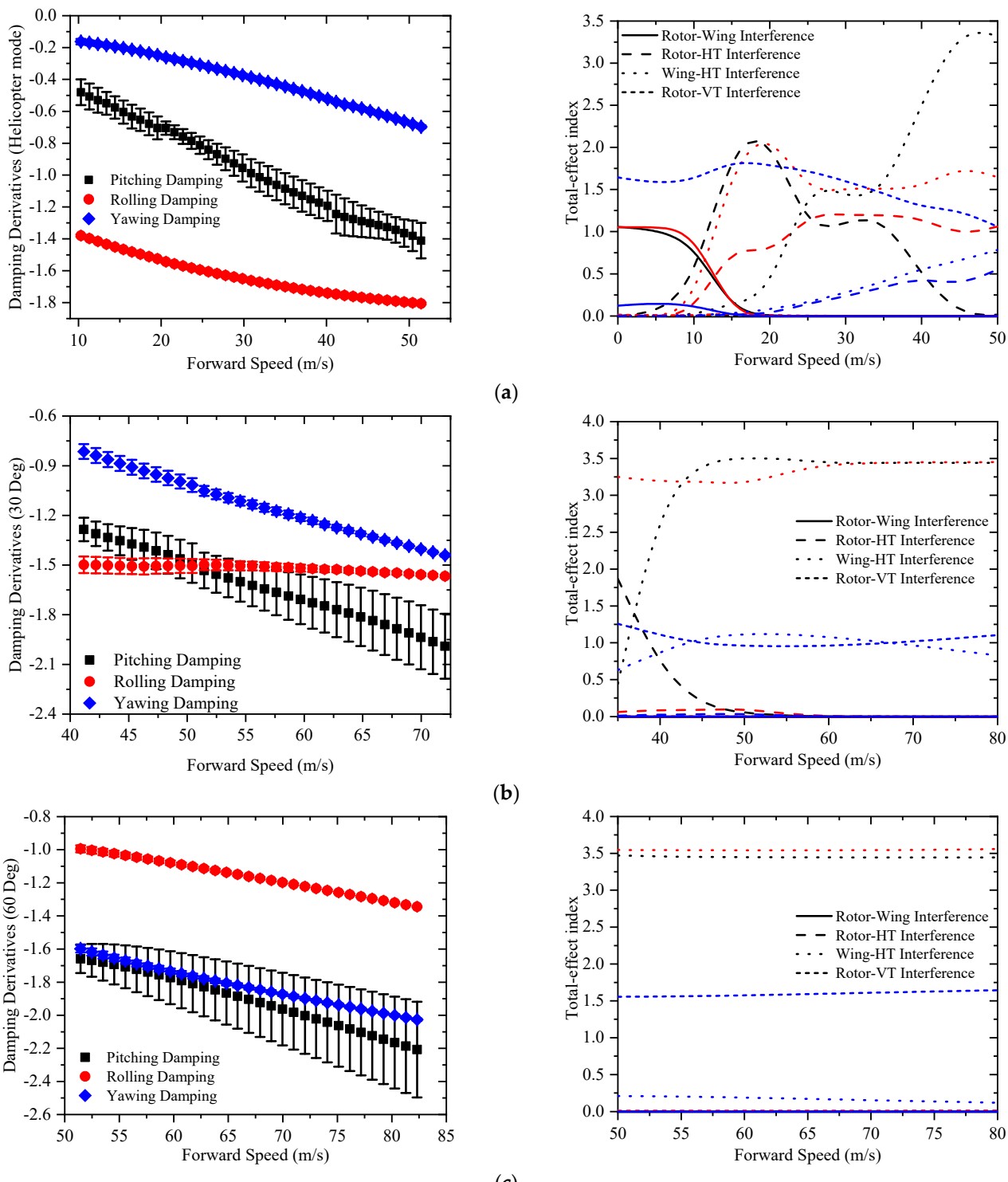

**Figure 7.** *Cont.*

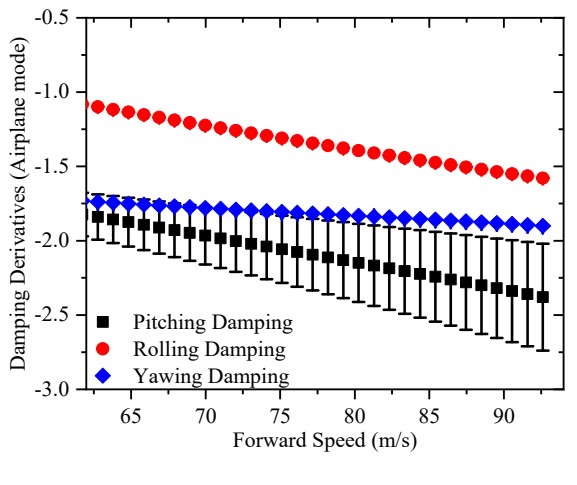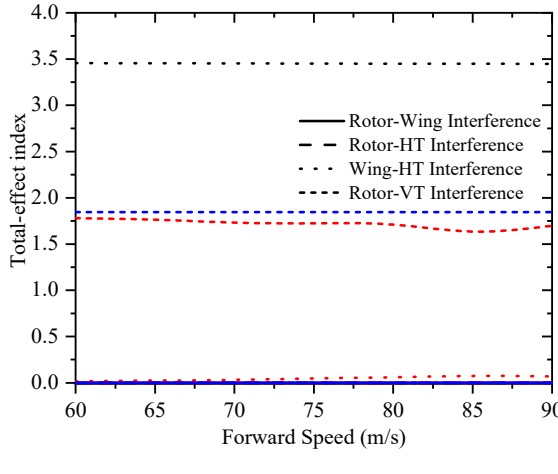

**(d)**

**Figure 7.** Angular damping derivatives. (**a**) Helicopter mode; (**b**) Conversion mode (30 Deg); (**c**) Conversion mode (60 Deg); (**d**) Airplane mode.

## 5. Conclusions

This article utilises the uncertainty quantification method to investigate the aerodynamic interference effect on the handling qualities of the tiltrotor aircraft, namely the eigenvalue and bandwidth and phase delay. Furthermore, the sensitivity analyses on the trim results and stability and controllability derivatives are also performed to illustrate the aerodynamic interference effect on the tiltrotor flight dynamics characteristics. The main conclusions from the current work are as follows:

1.  The quantification results indicate that aerodynamic uncertainties significantly alter the handling quality rating during helicopter mode and conversion and airplane modes with high forward speed. Thus, a simplified aerodynamic modelling method can be adopted in other flight ranges to improve the modelling efficiency.
2.  The trim results indicate that the influence of the aerodynamic interference on the trim characteristics is reduced as the nacelle angle increases. When the tiltrotor is in helicopter mode with higher forward speed, this interference may lead the horizontal tail to stall condition and degrade the handling qualities.
3.  The aerodynamic interference significantly influences the velocity stability in the helicopter mode and the incidence stability in the conversion and fixed-wing aircraft modes with higher forward speed. The handling qualities are significantly affected by the aerodynamics at these two flight ranges.

**Author Contributions:** Conceptualization, Y.Y. and D.A.; methodology, Y.Y.; software, Y.Y.; validation, Y.Y. and D.T.; formal analysis, D.T.; investigation, Y.Y.; resources, D.A.; data curation, Y.Y.; writing—original draft preparation, Y.Y..; writing—review and editing, D.T. & D.A.; visualization, Y.Y.; supervision, D.A. & D.T.; project administration, D.T. and D.A.; funding acquisition, D.T. and D.A. All authors have read and agreed to the published version of the manuscript.

**Funding:** The financial support of the EPSRC project MENtOR: Methods and Experiments for NOvel Rotorcraft EP/S013814/1, is gratefully acknowledged.

**Informed Consent Statement:** Not applicable.

**Data Availability Statement:** Not applicable.

**Conflicts of Interest:** The authors declare no conflict of interest.

## Nomenclature

| | |
|---|---|
| $E$ | statistical moments |
| $g_p$ | probability density function |
| $S_{Actuator}$ | actuator transfer function |
| $S_{control}$ | control mechanism transfer function |
| $t$ | response time (s) |
| $\boldsymbol{u}$ | control vector |
| $U,V,W$ | translational velocities in local axes (m/s) |
| $v_i$ | induced velocity on the rotor disc (m/s) |
| $\boldsymbol{x}$ | state vector |
| $X,Y,Z$ | force component in local axes (N) |
| $\alpha$ | angle of attack (Deg) |
| $\beta_m$ | nacelle incidence angle (Deg) |
| $\omega_{Bw\theta}$ | bandwidth (rad/s) |
| $\delta$ | aerodynamic uncertainty factor |
| $\tau_{p\theta}$ | phase delay (s) |
| *Subscripts* | |
| *int* | interference |
| *ht* | horizontal tail |
| *rh* | rotor-horizontal tail interference |
| *rv* | rotor-vertical tail interference |
| *rw* | rotor-wing interference |
| *vt* | vertical tail |
| *w* | wing |
| *wh* | wing-horizontal tail interference |

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
