# Peer review of "Aerodynamic Uncertainty Quantification for Tiltrotor Aircraft"

_aerospace, doi:10.3390/aerospace9050271_

Round 1
Reviewer 1 Report
The tiltrotor has unique flight dynamics due to the aerodynamic interference characteristics. Multiple aerdoyanmics calculation approaches such as CFD are time-consuming and the obtained results are generally varied from each other. In this paper, the uncertainty quantification method will be utilised in this research to identify the aerodynamic inaccuracy effect on the handling qualities of the tiltrotor aircraft. The method is newly applied on the analysis of tiltrotor handling quality and the obtained results are new. The writing is well strcutred with no visible flaw. The results are discussed in detail and in depth. I recommend direct acceptance of this paper to be published in the journal.
Author Response
Thank you for your comment.
Reviewer 2 Report
The authors have performed a study utilizing the uncertainty quantification method to investigate the aerodynamic interference effect on the handling qualities of the tiltrotor aircraft along with a sensitivity analysis on the trim
results and main derivatives to illustrate the effect on the tiltrotor flight dynamics characteristics. I find the work done really good thus, I would recommend the manuscript to be accepted with minor revision. I only have few concerns before making this work available to the scientific community.
- At the end of page 5 the authors state 'It should be mentioned that the wing-vertical tail interference is ignored in the tiltrotor aircraft as their relative position suggests that this interference cannot
alter the overall flight dynamics to a large extent.' Is it possible to provide a reference or some preliminary analysis results to support the choice done? - Can the authors add some more details on the choice of the thresholds values to better clarify whether the setting of these values can influence the results obtained?
Author Response
Thank you for your comments.
As for the first question:
Lots of modelling and wind tunnel experiments indicate that this type of aerodynamics interaction would not influence the overall flight dynamics characteristics. In order to support this statement, we have added three citations to the manuscript. The revised section is highlighted in blue in the manuscript.
As for the second question:
The UQ threshold considers the performance characteristics and the handling qualities of the baseline calculation results. In this article, the ADS-33 is utilised here as the basis for deciding the threshold for the UQ process. On the other hand, the handling qualities are also determined by the flight speed and nacelle incidence angle. Thus, with the aim of demonstrating the aerodynamics effect more accurate, we alter the threshold through the level defined in the ADS-33 along with the flight states. The revised section is highlighted in red in the manuscript.